# Nearshore Contamination Monitoring in Sandy Soils Using Polymer Optical Fibre Bragg Grating Sensing Systems

**DOI:** 10.3390/s22145213

**Published:** 2022-07-12

**Authors:** Sina Fadaie, Moura Mehravar, David John Webb, Wei Zhang

**Affiliations:** 1College of Engineering and Physical Sciences, Aston University, Birmingham B4 7ET, UK; 200208408@aston.ac.uk (S.F.); m.mehravar@aston.ac.uk (M.M.); 2School of Optoelectronic Engineering, Qilu University of Technology, Jinan 250353, China; zhang.wei@qlu.edu.cn

**Keywords:** polymer optical fibre Bragg grating, health monitoring, nearshore geo-structures, contaminated soil

## Abstract

Civil engineering assets and geo-structures continually deteriorate during their lifetime, particularly in harsh environments that may be contaminated with corrosive substances. However, efficient and constant structural health monitoring and accurate prediction of the service-life of these assets can help to ensure their safety, performance, and health conditions and enable proper maintenance and rehabilitation. Nowadays, many of the largest cities throughout the world are situated in coastal zones, leading to a dramatic increase in the construction of nearshore geo-structures/infrastructures which are vulnerable to corrosion attacks resulting from salinity contamination. Additionally, seawater intrusion can threaten the quality and the sustainability of fresh groundwater resources, which are a crucial resource in coastal areas. To address these issues, detection of salinity in soil utilizing a novel polymer optical fibre Bragg grating (POFBG) sensor was investigated in this research. Experiments were carried out at different soil water contents with different salinities to assess the sensor’s response in a representative soil environment. The sensitivity of the POFBG sensor to salinity concentrations in water and soil environment is estimated as 58 ± 2 pm/%. The average standard error value in salinity is calculated as 0.43% for the samples with different soil water contents. The results demonstrate that the sensor is a promising and practical tool for the measurement and monitoring with high precision of salinity contamination in soil.

## 1. Introduction

Urban geo-structures and infrastructures such as foundations, retaining structures, buried utilities, road subbase and railway foundations play a critical role in the economy of any country throughout the world. Due to climate change, human activities, urbanization and the inherent uncertainty and complexity of geological conditions, stability is one of the major concerns during the design and construction phases of these geo-structures and infrastructures [1]. Without effective geotechnical monitoring procedures, including precise measurement of physical parameters of the ground (e.g., strain, water content, salinity contamination and temperature), unexpected total or partial failure of the ground may lead to catastrophic injuries, fatalities and significant financial loss. Climate change is one of the significant global catastrophic risks and is expected to intensify, which will have a negative impact on the stability and serviceability of nearshore geo-structures [2]. Among many issues linked to climate change, seawater intrusion has been widely accepted as one of the most challenging environmental problems throughout the nearshore environment [3]. Apart from the climate change impact on seawater intrusion, there are different factors which can cause seawater intrusion including pumping of groundwater, seasonal fluctuations, urban development (infrastructure, geo-structures, buildings, pavement), agricultural practices, natural causes, etc. [3].

Coastal regions have been widely accepted as economically and environmentally vibrant areas that support a variety of living things and diverse ecosystems. Coastal inhabitants and industries need a reliable source of freshwater. Due to increased utilization of groundwater in coastal areas, it has been recognized that groundwater supplies are vulnerable to overuse and contamination. Contamination of groundwater resources causes degradation of drinking-water supplies [4]. This challenge is fundamentally that of seawater intrusion into the freshwater [5], which reduces fresh groundwater supply when concentrations of dissolved ions surpass the drinking-water standards [3]. In other words, seawater intrusion threatens coastal freshwater resources globally, rendering groundwater non-drinkable [6,7,8] To clarify the effect of seawater intrusion in such cases, it should be mentioned that the saltwater and freshwater zones in coastal regions are separated by a zone of dispersion/transition within which there is mixing between freshwater and saltwater.

On the other hand, seawater has been widely accepted as a corrosive environment that exerts influence on infrastructure particularly those which are built in nearshore regions [9]. Nearshore geo-structures and infrastructures are mainly built of either steel or reinforced concrete as two fundamental materials in civil engineering. In this respect, carbon steel is a very common material used for coastal structures due to its functional properties such as high strength, ease of welding, and low cost compared with other metallic or plastic materials. However, due to its limited resistance to marine corrosive substances, it should be constantly monitored, maintained, and protected [10]. Furthermore, one of the most fundamental parts of a structure is the foundation due to its key role in structural stability [11]. Foundations of structures located at nearshore sites are exposed to corrosion due to the saline water. Soil is an essential constituent of construction activities; it is affected by seawater intrusion in nearshore areas causing disturbances in superstructures and resulting in soil salinization. The soil-salinity interactions can dramatically change the soil mechanical behaviour and also lead to various geotechnical problems. Saline water leaching and salt concentration in the ground increase the soil compressibility, permeability, and void ratio [12]. Moreover, in wet seasons, as a result of seawater intrusion, salts precipitate in the ground layers which in dry seasons leads to the formation of salt crusts. Once the crust becomes wet, the salts dissolve, and the soil is prone to catastrophic failure due to the wet-dry cycling impact [12,13]. In general, this process results in a loose structured soil characterized by low bearing capacity and with high settlement [14]. This poor soil condition leads to foundation failures and associated structural damage to infrastructure which are attributed to the susceptibility of the salts to dissolve, the soil to exhibit large volume changes upon wetting, and low bearing capacity [14]. There is clear evidence confirming the detrimental effect of salt concentrations on soil strength leading to decreasing the bearing capacity of foundations [15,16,17,18,19]. Therefore, there is an essential need to monitor the salinity concentration in soil environments exposed to constant seawater intrusion [20].

Moreover, one the most important aspects in the planning process in infrastructure projects is to consider methods of prevention and protection to avoid the interaction between the structures and contaminated materials such as saline soils. In spite of widespread seawater contamination of civil engineering infrastructures in nearshore environments, utilization of sensing systems has demonstrated that maintaining suitable performance in infrastructure over a long lifetime is not an impossible goal [2]. In recent decades, there has been an increasing number of studies on the application of fibre optic sensors to measure salinity concentration, though none of these studies deal with measurements in partially saturated soil as is the case in this paper. This previous work is discussed in Section 6.

In this paper, the sensitivity of a fibre Bragg grating (FBG) sensor fabricated in polymer optical fibre (POF) to soil salinity was investigated under a wide range of concentrations of salinity at various soil water contents less than the saturation degree of the soil. Sensitivity is here defined as the rate of change of the Bragg wavelength with respect to the percentage of NaCl dissolved in the water, defined as follows:(1)$$\mathrm{Percentage}=\frac{m_{NaCl}}{\left(m_{water}+m_{NaCl}\right)}\times 100.\;$$
where $m_{NaCl}$ is the mass of salt and $m_{water}$ is the mass of water in which the salt is dissolved.

The effect of soil temperature is also considered, and a temperature correction factor determined to remove the temperature cross-sensitivity effect. The results illustrate that the sensor can be a practically useful tool for the measurement and monitoring of soil salinity with high precision in civil engineering applications.

## 2. Sensor Principle and Fabrication

The increasing worldwide attention being paid to improving the resilience of geo-structures, infrastructures, and communities against natural disasters has opened up novel and challenging monitoring applications for fibre optic technology. In countless such advanced applications where miniaturization, sensitivity, and remote measurements are vital, optical fibre-based sensing techniques can provide solutions [21]. Among different types of fibre-optic sensors, fibre Bragg grating-based sensors are extensively used in geo-structure health monitoring due to their outstanding advantages including multiplexing capabilities, small size, electromagnetic interference immunity, remote operation as well as high resolution and absolute-measurement [22,23,24,25,26]. A fibre Bragg grating (FBG) takes the form of a periodic refractive index change along a short section (~cm) of the core of the optical fibre, typically produced by exposing the fibre to a spatially varying pattern of ultraviolet light [27]. Figure 1 shows the structure of a fibre Bragg grating and the approach used in this work to monitor the light back reflected from the grating.

When light of a broad spectrum is guided along the core of the fibre to the grating, light close to one wavelength known as the Bragg wavelength, is reflected back while the rest of the spectrum passes through unaffected (see Figure 1) [28]. The reflected Bragg wavelength, $\lambda_{B}$, depends on two parameters: the effective core refractive index of the fibre, $n_{e}$; and the grating period, Λ, as expressed by Equation (2) [29]:
(2)$$\lambda_{B}=2n_{e}\Lambda$$

Exposure of the FBG to changing strain and temperature causes variations in $n_{e}$ and Λ, and therefore, in the Bragg wavelength [29]. FBGs are increasingly being used across a very wide range of applications due to two main factors. Firstly, they possess a number of attractive features: (i) optical fibre is of low loss, enabling passive operation over long distances; (ii) the fibre is light in weight and of small diameter, facilitating its use in weight-critical applications and allowing it to be embedded within structures (or used within the body) in a minimally invasive fashion; (iii) being constructed from a dielectric material, FBG sensors are inherently immune to electromagnetic interference; (iv) techniques exist that allow multiple FBG sensors (in some cases many hundreds) to be multiplexed along a single fibre. Secondly, via appropriate sensor packaging, many other measurands can be transduced into a strain signal, so that FBGs are also able to sense ultrasound, force, acceleration, pressure, inclination, electric field and chemical concentration. Several review papers discuss these features in detail (see for example [27,30]) and describe the very wide range of applications that have been developed in, for example, harsh environments [31], the oil and gas industry [32], medicine [33], structural health monitoring [26], aerospace [34], biochemical sensing [35], and geo-structure monitoring [22,23,24,25,26].

Whilst the majority of work on FBG sensors has concerned devices fabricated in silica fibre, the technology has also been extended to polymer optical fibre (POF), where in addition to sensitivity to strain and temperature, the Bragg wavelength can also depend on the relative humidity surrounding the fibre [36], depending on the particular polymer used for the fibre. Poly (methyl methacrylate) (PMMA) is a common material from which POF is manufactured and it has an affinity for water, the absorption of which leads to a swelling of the fibre and an increase in the fibre’s refractive index, both leading to an increase in the Bragg wavelength. The water absorption is a reversible equilibrium process, such that the amount of water absorbed by the fibre is determined by the relative humidity of the environment around the fibre [37]. Furthermore, the water affinity of PMMA can also be used to monitor the concentration of a solute dissolved in water, with the amount of water absorbed by the fibre determined by the concentration of the surrounding liquid. This has been demonstrated with both sugar and, most relevant to the current work, salt [38].

Given the sensitivity of POFBGs to salt solution, in this work we have investigated whether a POFBG based sensor can be utilized to monitor the concentration of salt water in the unsaturated soil environment, where the POFBG itself is not in direct contact with any liquid. For this research, a POFBG has been inscribed in a few cm length of single mode, step-index, PMMA-based fibre using a HeCd laser and phase mask—please see [36] for further information. Figure 2 illustrates the reflection spectrum of the POFBG. The fibre diameter was approximately 95 µm, and the fibre was glued to a silica SMF pigtail. The sensor was also glued on either side of the FBG (Figure 3a) under a small amount of tension to an invar plate to prevent the fibre from experiencing significant strain induced shifts in the Bragg wavelength. The fibre on its invar plate was placed inside a metal tube (diameter = 8 mm) to protect it against the harsh soil environment. One end of the tube is covered by a metallic mesh to prevent sand particles from entering the sensor housing. Due to the metallic mesh holes, free circulation of air can occur, allowing an equilibrium to be established between the sensing fibre and surrounding sand (Figure 3b), but as noted earlier, there is no direct contact between the POFBG and the soil. The Bragg wavelength of the POFBG is measured by illuminating it using a broad-band source (Agilent 83437A) emitting around 50 μW in the 1550 nm spectral region, with the reflected signal from the POFBG being monitored using an I-Mon spectrometer from Ibsen Photonics—this unit is able to provide 512 measurement points across the spectral region 1510–1595 nm [23].

A potential disadvantage of such sensors is the cross-sensitivity to temperature. In this study, temperature sensitivity was dealt with by separately monitoring the soil temperature and then applying a temperature correction factor to the wavelengths [39]. In a practical system, temperature cross-sensitivity might be most effectively dealt with by mounting a silica fibre-based FBG in the sensor, which would respond to temperature but would not be affected by changes in water content or salinity.

## 3. Sensitivity Assessment of Polymer Sensor

Prior to undertaking experiments with the sensor in soil, we characterized the POFBG’s response to temperature and humidity. This information is useful because, unlike with silica FBGs, the sensitivity of polymer gratings can be affected by the polymerization process used to create the fibre preform.

### 3.1. Experimental Arrangement of Sensitivity Assessment

The temperature and humidity responses of the sensor were calibrated by housing the sensor inside an environmental chamber (Binder KBF 115). The experimental arrangement is schematically illustrated in Figure 4.

### 3.2. Temperature Sensitivity of Polymer Sensor

The temperature response of the sensor was assessed under different levels of constant relative humidity (RH): 40%, 50%, 60% and 70%. The changes in wavelength were observed and recorded at increasing temperature increments of 5 °C from 20 °C to 40 °C. As the humidity of the lab environment was usually in the range 38% RH to 45% RH, a more detailed assessment of the temperature sensitivity of the sensor was repeated at 40% RH with increments of 1 °C from 18 °C to 22 °C, which covered the range of temperature observed in the soil environment during the experiments. Figure 5 shows the sensitivity to temperature changes at various levels of relative humidity.

As can be seen, the polymer sensor’s wavelength has a declining trend with increasing temperature, due to the negative thermo-optic contribution of PMMA. In addition, as is evident from the results in Figure 5, increasing the humidity leads to an increase in the wavelength. The response of the polymer sensor to temperature reveals similar trends at different levels of humidity. The linear regressions illustrated in Figure 5 provide the temperature sensitivities of the POFBG at different constant humidities (see Table 1).

According to the data drawn from the linear regression, a temperature correction factor of 0.12 nm per °C was determined and applied to the results presented later, where the temperature of the soil and laboratory environment was constantly monitored by a mercury-in-glass thermometer. According to the data drawn from the linear regression, as an example, the temperature sensitivity of the sensor is 114.40 ± 0.99 pm/°C at 40% RH, which is in good agreement with the correction factor calculated by [23]. This, high value of sensitivity seems anomalous at first glance—much higher than common values reported for POFBGs (e.g., 55 ± 3 pm/°C at a constant 50% RH) [40]—it is, however, expected, given the construction of the sensor. In PMMA-based optical fibre Bragg gratings, the temperature response is composed of thermal expansion and the thermo-optic effect of the fibre. Thermal expansion makes a positive contribution while the thermo-optic coefficient makes a negative contribution to the Bragg wavelength shift of the POFBGs. Therefore, they cancel out each other to some extent. In our case, the sensitivity is much higher than the sensitivity of a free fibre because the fibre has been deliberately fixed under tension to its supporting invar plate, so there is no contribution to the sensitivity from thermal expansion [39].

### 3.3. Humidity Sensitivity of Polymer Sensor

To quantify the humidity sensitivity of the polymer sensor, experiments were carried out at two different fixed temperatures (20, 25 °C) while the humidity was varied from 40 to 70% RH with a 10% RH increment for each step. These two temperatures have been selected as the temperature in the soil varied from 20 to 22 °C for each sample in this research. Moreover, the lab temperature changed from 20 to 22.5 °C during the tests. According to [40], the response time of a POFBG sensor to a step change in humidity can be around 30 min, though it strongly depends on the fibre diameter and the detailed material properties. However, in this study we waited 1 h after each humidity change before recording results to ensure the system was in equilibrium. The results of this characterisation are presented in Figure 6.

As is evident, there is effectively a linear relationship between humidity and the sensor’s wavelength. Table 2 presents the sensitivities to humidity obtained at the two temperatures, showing they are consistent.

## 4. Experimental Design

All soil samples used in this study were Leighton Buzzard sand, with a specific gravity of 2.66 and a nominal effective size of 0.63–0.85 mm [23]. A particle size distribution analysis was performed in accordance with the British Standard Institution [41], and the soil particle size distribution curve is illustrated in Figure 7.

The soil samples were oven dried at 105 °C for 24 h and then kept in the lab environment for at least 24 h to be in equilibrium state with the lab environmental conditions. To generate predefined gravimetric water contents for the samples, the required amount of distilled water was calculated by Equation (3):(3)$$\omega =\frac{m_{w}}{m_{s}}.\;$$
where ω is soil water content, $m_{w}$ is the mass of water, and $m_{s}$ is the mass of soil solid particles in the sample. In Equation (3), $m_{s}$ was considered as the solid mass after oven-drying which comprised the combined mass of soil and salt. To prepare pre-defined salinity concentrations, the required mass of NaCl was also added to the calculated water content. Saline water was added using a pipette to the dry soil, and the soil–saline water mixture was mixed for a duration of 3 min using an electrical stand mixer. The amount of dry soil was constant for each set of experiments. A standard proctor mould of volume 1000 cm^3^ was employed to compact the soil samples in three equal layers according to the BS 1377-4 standard [42]. The dry density and porosity of the soil are stated in Table 3. The dry density ($\rho_{d}$) for saltless soil can be calculated by the following equation:(4)$$\rho_{d}=\frac{\rho }{1+\omega }$$
where ρ is density of the compacted sample. It should be noted that the dry density is overestimated in this research due to the consideration of the dissolved salt as soil solids —in saline samples when the sample is oven-dried, the water is evaporated, whereas the salt remains with the dry soil solids [43]. In this study, each test was repeated three times to ensure the consistency of the experiments and their results.

The experiments were carried out at 5, 10 and 15% soil water content (SWC) with different salinity concentrations. These specific values of soil water content were selected because the samples were compacted up to near saturation condition [23]. The degree of saturation for the samples with different values of water content was calculated by Equation (5), a well-known formulation in soil mechanics [44]:
(5)$$S_{r}\times e=\omega \times G_{s}$$
where ω defines the soil water content, e is soil void ratio, $S_{r}$ is the degree of saturation, and $G_{s}$ defines the specific gravity of the soil.

As can be seen in Figure 8, it would be impractical to carry out the standard proctor test above 18% water content for the compacted soil because the sample is approaching full saturation. The sample preparation data are summarized in Table 4.

The POFBG sensor was inserted into the soil sample to a depth of 5 cm. A mercury-in-glass thermometer was used to monitor the temperature changes in the soil environment, which were in the range of 20 to 22 °C during the experiments due to room temperature fluctuations. In each test, the soil temperature was recorded, and the POFBG’s temperature correction factor was applied to the results. Moreover, the humidity of the laboratory environment was constantly monitored during the experiments and varied from 38% RH to 45% RH. The arrangement of the experiment to monitor the salt concentrations in the soil is schematically illustrated in Figure 9.

In addition, it is worth stating that even when the sensor is inserted directly into liquid, the construction of the sensor (the fact that the tube is sealed at the top) does not allow liquid to enter through the mesh and come into contact with the fibre. Therefore, the response of the sensor is via an equilibrium process whereby the humidity of the air in the sensor chamber comes into equilibrium with the liquid or soil sample and then the fibre comes into equilibrium with the air in the sensor.

## 5. Results and Discussion

### 5.1. Sensor’s Sensitivity in Saline Water

In order to assess the sensitivity of the sensor to NaCl concentration, the sensor was immersed in NaCl solutions at different concentrations. The experiments were carried out at six different NaCl weight percentages: 0, 5, 10, 15, 20 and 25% in pure distilled water. Figure 10 presents the polymer sensor’s response to salt concentration changes.

It may be seen that following initial immersion in pure water, there is a significant increase in the Bragg wavelength as the fibre absorbs water. From that point, increasing the salt concentration leads to a negative wavelength shift as osmotic pressure draws water out of the sensor cavity and out of the polymer forming the fibre itself.

Using the average of the last 10 min of recorded wavelengths at each concentration, a linear regression was produced and is illustrated in Figure 11.

It can be observed that there is a linear relationship between NaCl percentage concentration and the sensor’s wavelength which can be expressed by Equation (6):(6)$$\lambda =-0.0573S_{c}+1537.5$$
where 𝜆 and $S_{c}$ are the sensor’s wavelength in nm and the salt concentration, respectively. The sensitivity of the sensor was determined via linear regression to be 57.3 ± 2.0 pm/%.

### 5.2. Assessment of POFBG’s Reversible Behaviour

In order to assess the reversible nature of the measurement process of NaCl concentration using the sensor, the polymer sensor was placed into solutions with various percentages of salinity. As can be seen in Figure 12, when the salt concentration is minimum (0%) in pure distilled water, the wavelength shift is maximum. With an increase in concentration, wavelengths decrease as illustrated in Figure 12. After a steady wavelength was obtained in 20% NaCl, the environment was immediately switched to the former stage (5% NaCl) and then pure distilled water for the final stage.

Figure 12 shows that the recorded wavelengths are consistent for the same value of NaCl (5%). Using the average wavelength measurement over the last 10 min recorded in each phase of the experiment, the difference between the two measurements at 5% NaCl solution was 15 pm, whilst for pure distilled water it was 2.5 pm. These extremely small differences confirm the repeatability of the sensing process and probably mainly arise due to the temperature variation in the solution over the period of the experiment (about 7 h).

### 5.3. Sensor Response to Salt Concentration in Contaminated Soil

In order to assess the POFBG sensor’s response to changes in salinity concentrations in soil, samples with different NaCl percentages ranging from 5% to 20% were prepared in a controlled laboratory environment as described earlier. Initially, the SWC was specified at 15%, which is close to saturation condition. The effects of changes in salinity concentration in the defined compaction state on the sensor’s wavelength response were investigated. All the recorded wavelengths were determined from the average response of the sensor in the last 10 min for each sample. Figure 13 illustrates the sensor’s response to various NaCl percentages at 15% SWC. It is evident that the sensor is able to detect changes in salinity, and the spectrum becomes stable after about 1.5 h. The delayed reaction to the environment is due to the time required for the air space in the sensor metal tube and the polymer fibre itself to reach an equilibrium state with the soil mixture. Depending upon the intended application, the sensor can be redesigned to have a shorter response time–as short as a few minutes or less—by reducing the volume of airspace around the fibre [45].

Moreover, as may be seen in Figure 14, it is evident that there is a linear relationship between the salinity contamination in the soil (%) and the sensor’s wavelength (λ). The sensor’s sensitivity to salinity concentrations is estimated as 57.8 ± 2.0 pm/%, whilst the standard error value in salt concentration is calculated as 0.46%. This very low value of standard error clearly indicates the consistency of the data obtained throughout the experiments and demonstrates the reliability and accuracy of the sensor for salinity contamination measurement in soil environments. The sensitivity is also consistent with that obtained previously in salt solution (57.3 ± 2.0 pm/%).

Employing the same procedure, experiments were also carried out at 5% and 10% soil water content. Figure 15 illustrates the comparison of sensor response to salinity concentration in the soil environment at three different percentages of water content, 5%, 10% and 15%. As discussed, increasing salinity contamination causes the wavelengths to experience a declining trend. Additionally, it can be observed that, the greater the soil water content, the higher the wavelengths that can be obtained.

The experiments at each specific condition were repeated three times to ensure the consistency of the results. Averages taken from the three tests were used to determine the sensor response (see Table 5). It can be seen that the sensitivities of the sensor to the salinity contamination in the soil environment obtained at different water contents were almost identical. Moreover, the dependence on soil water content is a great deal smaller than the dependence on salt concentration, suggesting the sensor can be used to monitor salt concentration largely independently of the actual amount of water in the soil. This suggests the sensor is a promising tool to measure salinity contamination in sands under different conditions.

## 6. Discussion

Relevant previous work concerning salinity is summarized in Table 6.

All of the approaches described in Table 6 are concerned with measuring the salinity of water; ours is the only work that we are aware of that extends to investigating the monitoring of the salinity in partially saturated soil, which imposes specific requirements on the sensor design. Different methods have been used to deduce the salinity: several approaches have investigated refractive index as a proxy for salinity: one measures density, while most surround the sensor in a material which absorbs water and is sensitive to the amount of water (the water activity) in the sensor environment. This latter transduction mechanism is used by our sensor, but is intrinsic to our fibre material, rather than requiring a separate coating stage. Direct comparison in performance figures is not straightforward as different authors use different approaches to defining sensitivity; nevertheless, our sensitivity value of 58 pm/% is better than most reported. We would argue, though, that resolution is a more important performance metric than sensitivity, and it is serious omission that so few papers quote this measure.

It is important to note that, in common with our approach, none of these techniques provide a specific response to NaCl. In applications designed to provide an early warning of seawater encroaching on infrastructure, this is unlikely to be a serious deficiency. However, there will of course be other applications where it could be useful to differentiate between, for example, saltwater intrusion and heavy metal contamination.

## 7. Summary and Conclusions

Seawater intrusion is closely linked to the structural health of geo-structures and infrastructures in nearshore environments, and it also affects the quality and sustainability of fresh groundwater. Systematic monitoring can facilitate the prediction of these assets’ deterioration processes and protect the quality of ground fresh water. In this study, for the first time, a resilient and novel polymer optical fibre Bragg grating (POFBG) sensor was utilized to measure salinity contamination in soil. The work described in this paper focuses on the calibration of the POFBG sensor to detect salinity changes in the soil environment. For this purpose, firstly, the principle and fabrication techniques of the sensor were presented followed by the procedure of measurement using FBGs. The sensitivity and response of the sensor to both temperature and humidity were assessed. To evaluate the sensor performance in water environment, a wide range of salt concentrations was prepared. Then, the sensor was properly packaged and buried vertically in a set of sandy soil samples for the purpose of monitoring salt contamination. The samples were prepared with three different values of soil water content. The results revealed consistent sensitivities in different environments—saline-contaminated soil, and saline water. Multiple regression analysis revealed that the sensor can be utilized for salinity measurement in the desired circumstances. The conclusions drawn from the present research are as follows:
The sensor responds much more strongly to salt concentration than water content. The salt concentration can therefore be effectively determined without having to know the water content separately.The sensitivity of the POFBG sensor to salinity concentrations in water and soil environments was assessed and estimated as 58 ± 2 pm/%, as an average value. It is worth mentioning that the average standard error value in salinity was calculated as 0.43% for the samples with different soil water contents.The sensor was confirmed to respond reversibly to increasing and decreasing changes in salinity.POFBGs appear to be a promising tool for salinity contamination monitoring at different soil water contents. Therefore, the proposed sensor could be employed in geo-structures and infrastructures as an early-warning system in nearshore regions to detect any salinity variations caused by seawater intrusion. The utilization of the POFBG sensor could facilitate in situ measurement, allowing for the constant monitoring of the change of salt concentration at multiple points.Integrating the data gathered from this sensor with other key strength parameters of soil and infrastructures will provide an opportunity to dramatically decrease the detrimental impacts of salinity contamination in nearshore regions due to seawater intrusion.The insights gained from this study could also be applied to preserve a reliable source of freshwater for coastal inhabitants and industries.


However, further experimental investigations at laboratory and field scales are needed to ensure the sensor’s reliability and performance under different conditions, including different soil types with different compaction states. Moreover, further work is needed to establish the sensor’s long-term stability.

## Figures and Tables

**Figure 1 sensors-22-05213-f001:**
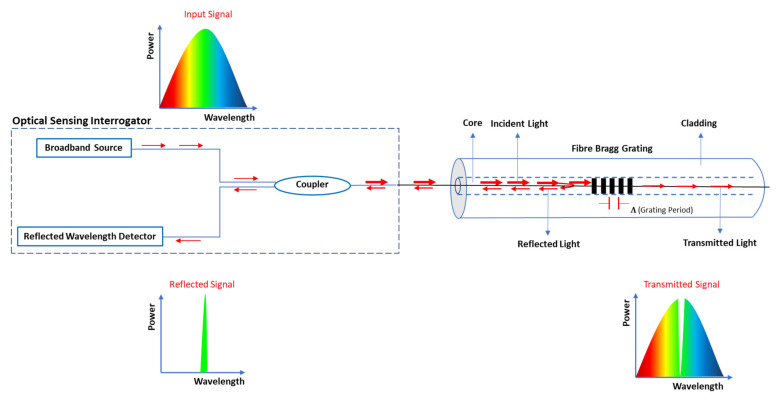
Structure of an optical fibre and scientific principle of FBG sensing technology.

**Figure 2 sensors-22-05213-f002:**
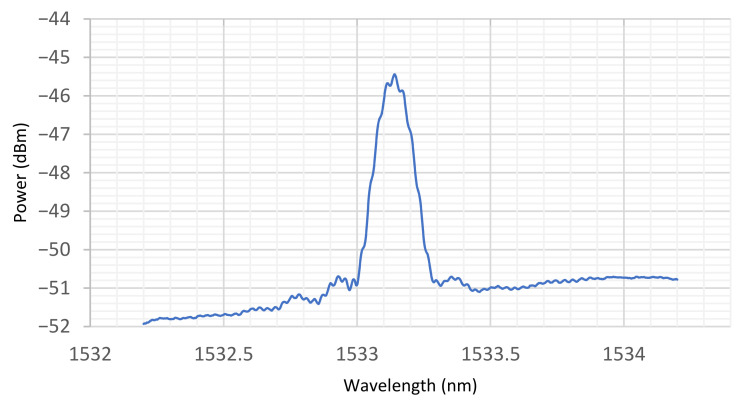
POFBG reflection spectrum.

**Figure 3 sensors-22-05213-f003:**
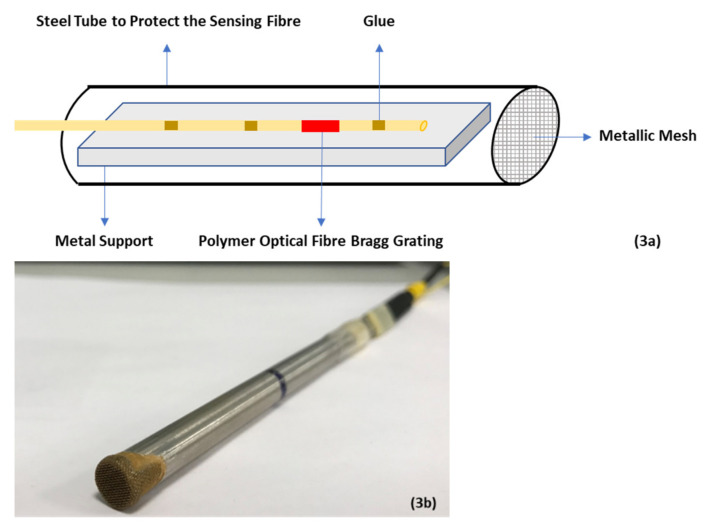
(**a**): Sensor construction, (**b**): The POFBG sensor packaging [23].

**Figure 4 sensors-22-05213-f004:**
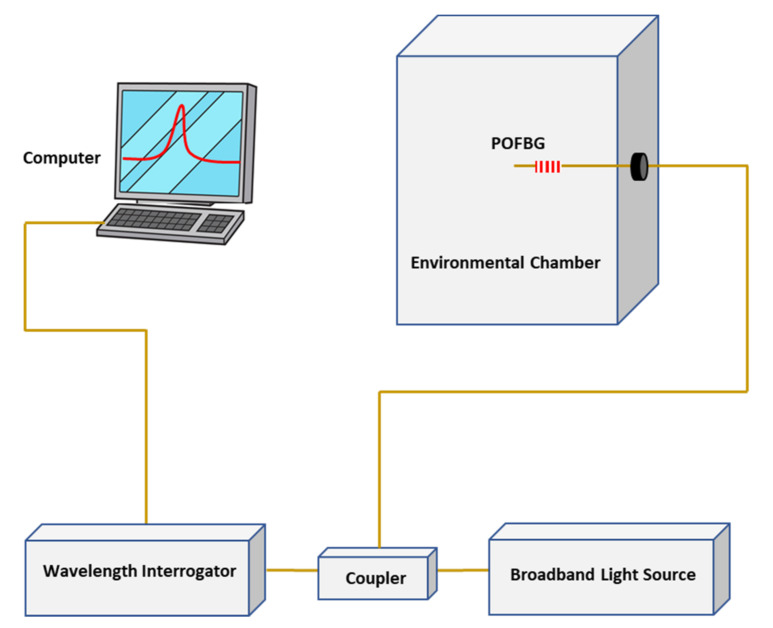
Experimental arrangement of the sensitivity assessment of the polymer sensor.

**Figure 5 sensors-22-05213-f005:**
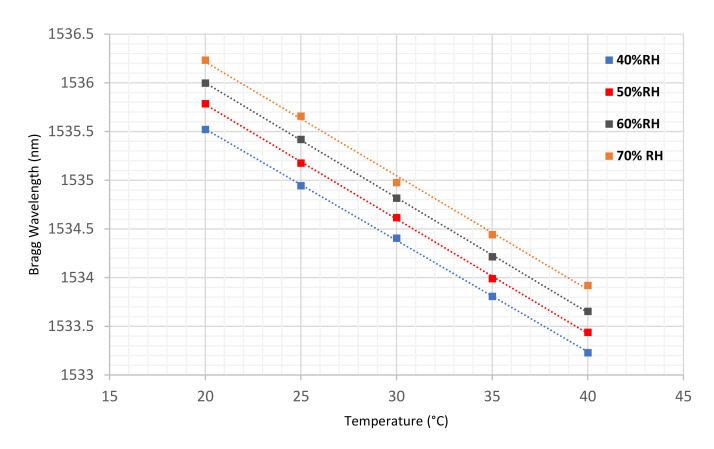
Comparison of the polymer sensor’s response to temperature changes at different levels of relative humidity.

**Figure 6 sensors-22-05213-f006:**
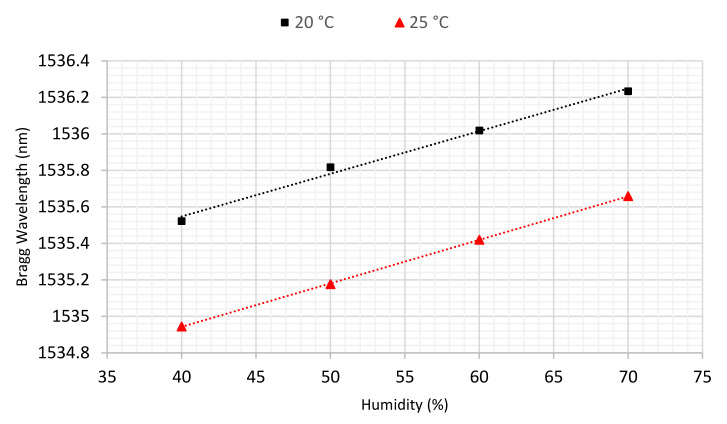
Humidity sensitivity of the polymer sensor at 20 and 25 °C.

**Figure 7 sensors-22-05213-f007:**
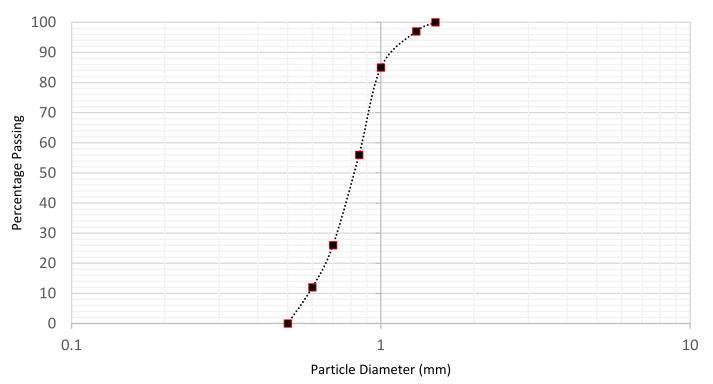
Soil particle size distribution curve.

**Figure 8 sensors-22-05213-f008:**
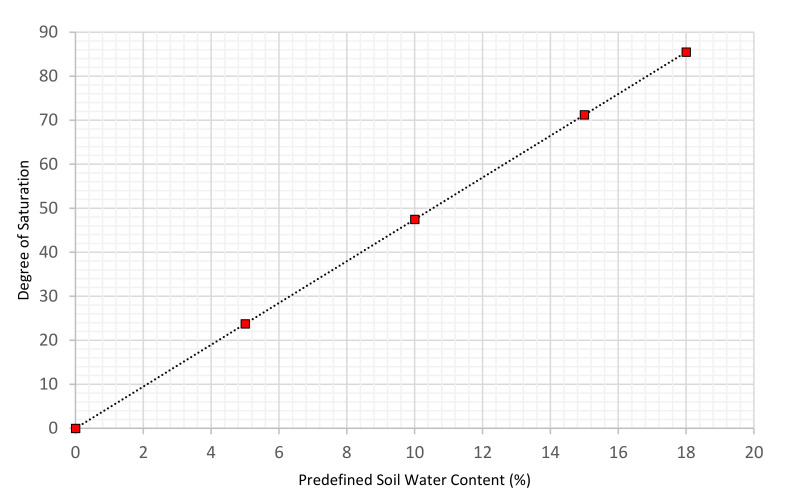
Saturation degree at different water content in compacted samples.

**Figure 9 sensors-22-05213-f009:**
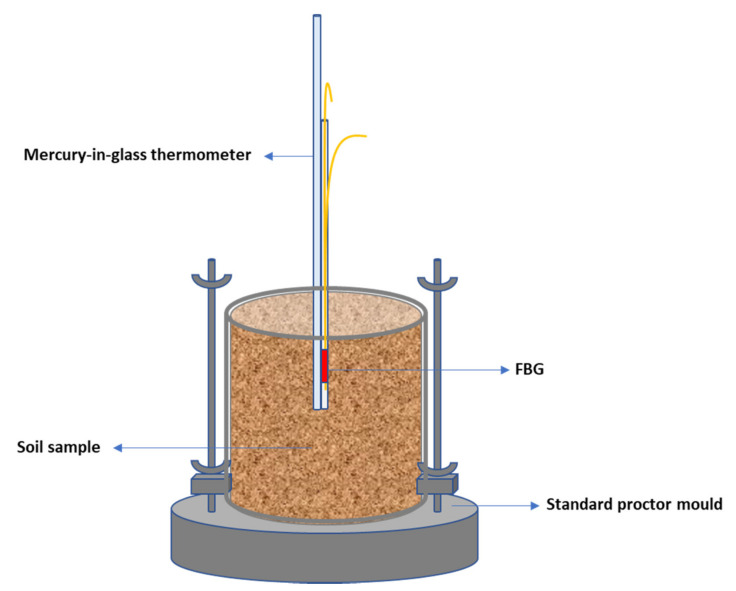
The arrangement of the experiment to monitor the salinity concentration in the soil.

**Figure 10 sensors-22-05213-f010:**
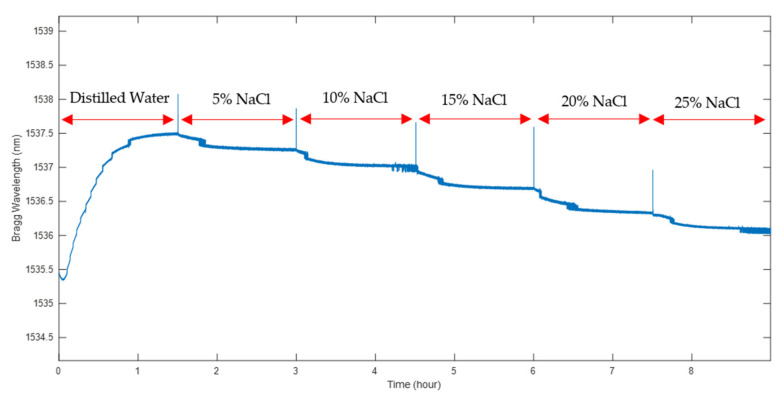
The polymer sensor’s sensitivity to salt concentration changes in water.

**Figure 11 sensors-22-05213-f011:**
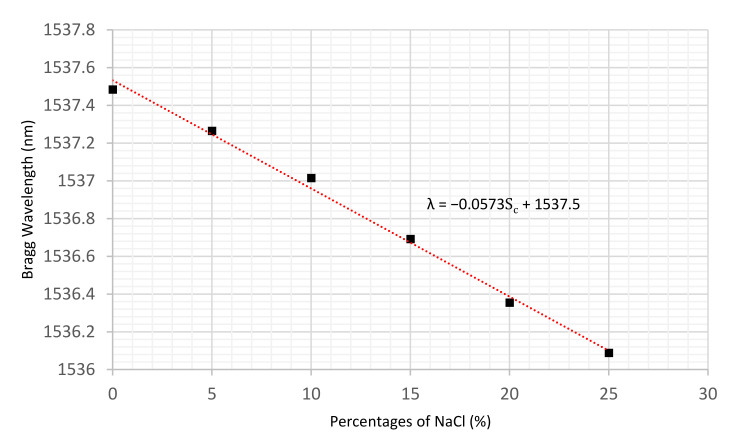
The linear regression of the polymer sensor’s response to salt concentration changes in water.

**Figure 12 sensors-22-05213-f012:**
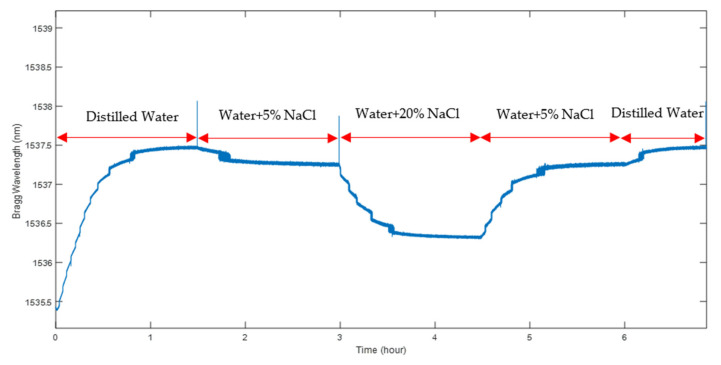
PMMA-based sensor assessment for reversible process in different NaCl concentrations.

**Figure 13 sensors-22-05213-f013:**
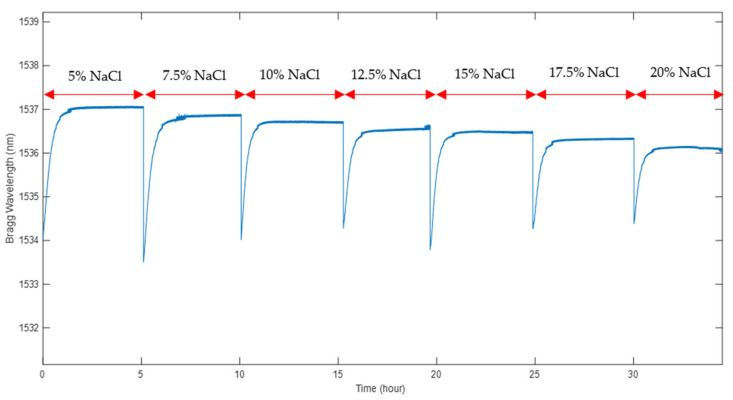
The sensor’s response to salinity variations at a constant SWC of 15%.

**Figure 14 sensors-22-05213-f014:**
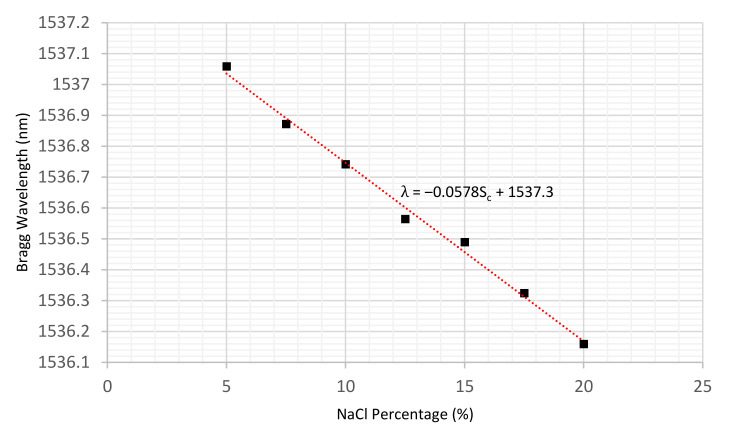
Linear relationship between salinity contamination and the sensor’s wavelength at a constant SWC of 15%.

**Figure 15 sensors-22-05213-f015:**
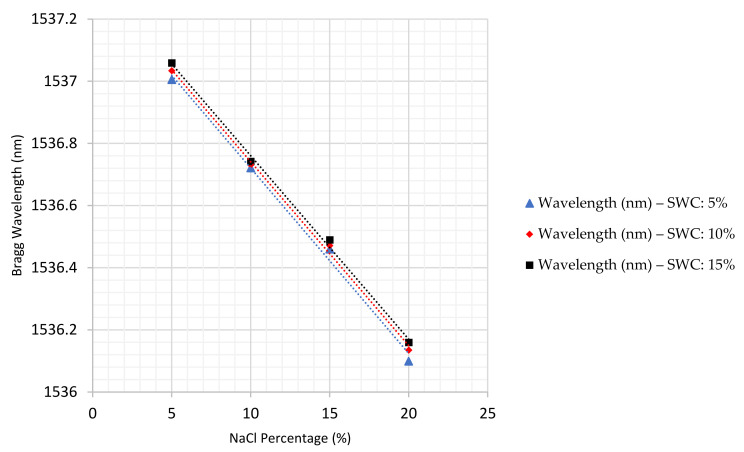
Comparison of salt concentrations at 5%, 10% and 15% soil water content.

**Table 1 sensors-22-05213-t001:** Temperature Sensitivity of POFBG (pm/°C) at different humidities (% RH).

Constant Humidity (%RH)	40	50	60	70
**Temperature Sensitivity** (**pm/°C**)	114.40	117.60	117.86	116.88
**Uncertainty** (**pm/°C**)	0.99	1.00	0.85	3.00

**Table 2 sensors-22-05213-t002:** Humidity sensitivity (pm/%RH) of the polymer sensor at 20 and 25 °C.

Temperature (°C)	20	25
**Humidity Sensitivity (pm/%RH)**	23.38	23.85
**Uncertainty (pm/%RH)**	1.47	0.15

**Table 3 sensors-22-05213-t003:** Soil dry density and porosity.

Test	$\mathrm{Dry}\;\mathrm{Density}\;(g/cm^{3})$	Porosity (%)
Standard Proctor Test	1.69	36

**Table 4 sensors-22-05213-t004:** Parameters used for the experiments carried out in this study.

Test	No. of Layers	No. of Blows Per Layer	Hammer Weight (kg)	Drop Height (cm)	Mould Volume (cm^3^)	Soil Water Content (%)	Predefined Salinity (%)
**Standard Proctor Compaction**	3	27	25	30	1000	5, 10 and 15	5, 7.5, 10, 12.5, 15, 17.5 and 20

**Table 5 sensors-22-05213-t005:** Sensitivity of POFBGs (pm/°C) to NaCl concentration at 5%, 10% and 15% soil water content.

Soil Water Content (% RH)	5	10	15
**Sensitivity to Salt Concentration** (**pm/%**)	58.6	58.8	57.8
**Uncertainty** (**pm/%**)	3.2	1.9	2.0
**Standard Error in Salinity** (**%**)	0.56	0.27	0.46

**Table 6 sensors-22-05213-t006:** Comparison of different optical fibre methods for salinity measurement.

Reference	Technique	Resolution/Sensitivity
[46]	Hydrogel-coated single-mode FBG sensor	Sensitivity: non-corrosive sensor: 2.1 pm/‰, corrosive sensor: 10.4 pm/‰
[47]	Fibre-optic refractive-index sensor based on surface plasmon resonance (SPR)	$\mathrm{Relation}\;\mathrm{between}\;\mathrm{refractive}\;\mathrm{index}\;\mathrm{and}\;\mathrm{salinity}\;\mathrm{of}\;∼3\times 10^{-7}$ RIU/ppm
[48]	A compact sensing head based on FBG technology	$\mathrm{Resolution}:\;0.2\%/\sqrt{\mathrm{Hz}}$ Sensitivity: 1.28 pm/%
[49]	Polyimide-coated photonic crystal fibre Sagnac interferometer based on the coating swelling induced radial pressure	Sensitivity: 0.742 nm/(mol/L)
[50]	Etched FBG coated with a layer of polyimide	Fundamental Mode Resonance Wavelength (FMRW) and Cladding Mode Resonance Wavelengths (CMRW) have the detection sensitivities of 15.407 and 125.92 nm/RIU for Refractive Index
[51]	Refractive Index using long-period fibre grating based Michelson interferometer	Chloride concentrations as low as 355 ppm
[52]	Long-period fibre grating functionalized with Layer-by-Layer Hydrogels	Sensitivity: 12.55 pm/%
[53]	Polymer-coated FBGs	0.0126 nm/M for KCL
[54]	FBG coated with lamellar polyimide	Sensitivity: 35.8 pm/%
[55]	Multiplexed polymer-coated FBG	Sensitivity: 1.6 pm/‰
[56]	Optical refraction method	Sensitivity: 7.8–9.3 µm/‰
[57]	Based on the measurement of the beam deviation due to the refractive angle change, which is nearly proportional to the salinity, by using a Charge Coupled Device (CCD)	Sensitivity: 16.2 µm/‰
[58]	Surface plasmon resonance method	Sensitivity: 200 pm/‰ in the range of 28–42%
[59]	Cascaded long-period fibre gratings (LPG)	Sensitivity: 10 pm/‰
[60]	Fabry-Perot interferometer (FPI)	Sensitivity: 8.1 pm/%
[61]	Salinity using Sagnac interferometer	Sensitivity: 1.95 nm/% Detection limit: 0.01%
[62]	Plastic optical fibre made of U-shaped and spiral probe salinity sensor (Changes of salinity in the water result in changes of light intensity in the transmitted fiber—the salinity value can be obtained by measuring the optical power of the final output light)	Sensitivity: U-type sensor is 0.042 mV/% and the spiral sensor is 0.013 mV/%

## Data Availability

The data presented in this study are available on request from the corresponding author.

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
