# Peer review of "Nearshore Contamination Monitoring in Sandy Soils Using Polymer Optical Fibre Bragg Grating Sensing Systems"

_sensors, 2022, doi:10.3390/s22145213_

Round 1

Reviewer 1 Report

I reviewed a manuscript entitled by “Nearshore contamination monitoring in sandy soils using polymer optical fibre bragg grating sensing systems.” The manuscript introduced a new fibre optic sensor to determine soil salinity. I think this is interesting and novel topic for many readers of the journal. The manuscript is very clear, easy to understand although the structure of the manuscript is a bit irregular, i.e., the third chapter “Sensitivity assessment of polymer sensor” includes all methods, results and discussion, followed by another “experimental design”. The weakness of the manuscript is that the authors tested this new method with only one salt solution and one soil. I think this is first introductive paper for the new technology so it might be okay, but I want the authors to discuss what kind of results are expected with other salt and soil if possible. Other salt may show different vapor pressure, and other soil may have different vapor diffusion speed. Other than that, I think this paper is ready to be published.

Reviewer 2 Report

This paper presents the study of the sensitivity a salinity sensor based in polymer optical fibers FBGs sensor for soil samples. A sensitivity to salinity of 58 pm/percentage was achieved and the viability of using this sensor to monitoring the salinity of soils was demonstrated. The approach used for design the sensor is interesting and the paper is well written and organized. However, there are some issue that need to be addressed.

-          First of all, the FBG reflection spectrum should be presented.

-          The humidity sensitivity of the sensor to humidity (Figure 5) was characterized in range of 40 to 70% RH with an increment od 10% RH, each translate in only 4 steps. Furthermore, in Figure 9, when the sensor is immersed in distilled water, appears to achieve a humidity higher than 70%. Is not possible to do the sensitivity to humidity characterization in larger range with more steps?

-          The sensor salinity sensitivity characterization was made in NaCl concentration range of 0 to 25% achieving a linear sensitivity. For higher concentration of NaCl the relation between the Bragg wavelength and concentration percentage remain linear? For instance, in [29] this relation is not linear.

-          The response of operation of the sensor for the salinity is related with the equilibrium between, the soil sample, air in the sensor and fiber. This equilibrium can be changed by any soluble substance. In this way, I don't think it will be possible to discriminate between the different possible contaminants in the soil. Furthermore, the temperature will also affect this equilibrium, the sensor characterization should be made for different temperatures, in both in both water and soil.

Reviewer 3 Report

Based on the demand for soil salinity monitoring in the field of structural health conditions, the manuscript proposed a sensor with a dynamically changing air chamber in the form of a tube. Through the wavelength tracking of the bare polymer fiber grating itself, without material modification, the relative humidity measurement is realized, and the sensor is also used to carry out salinity concentration monitoring. The experimental workload of the article is sufficient and innovative. But there are a few small problems with the manuscript

1 The measurement of salinity is calibrated by relative humidity. Whether the change in salinity is the only cause of the change in relative humidity? Whether there are other factors that may affect the accuracy of the sensor?

2 The response speed of the sensor seems to be very slow, as shown in Figures 9 and 12, the reaction to distilled water when reach equilibrium needs 1 hour. Is it because of the slow change of the monitoring environment itself, or the problem of the sensor mechanism, which is the dynamic balance of the air cavity is slow .

3. The problem of temperature correction. The thermometer and sensor shown in Figure 8 are in different areas, which will lead to temperature differences. Why not put them in the same  tube, or combine silicon-based optical fibers to achieve temperature calibration?

The author needs to explain these issues first.

Reviewer 4 Report

1. In the introduction, the author should discuss more applications of FBG.

2. The y-axis of figures 5, 10,11,12,13, and 14 need to be corrected. It should be replaced by wavelength to Bragg wavelength.

3. A table of similar types of studies should be prepared in the conclusion, as well as a discussion of why this technique is superior to others.

4. The sensitivity should be defined in the paper.

5. The author never explained why with an increase in salt concentration, Bragg wavelength changes. It is important for the author to explain the change in Bragg wavelength observed in the manuscript.

Round 2

Reviewer 2 Report

I am satisfied with the revisions. So, I recommend this paper for publication.